# Transcriptome analyses reveal differences in the response to temperature in Florida and Northern largemouth bass (*Micropterus spp*.) during early life stages

**Moisés A. Bernal**[1,2]*, **Gavin L. Aguilar**[3], **Josh Sakmar**[4], **Sebastian N. Politis**[5], **Savannah L. Oglesby**[3], **Allen Nicholls**[4], **Anita M. Kelly**[3], **Luke A. Roy**[3], **Ian A. E. Butts**[3]

1 Department of Biological Sciences, Auburn University, Auburn, AL, United States of America,
2 Smithsonian Tropical Research Institute, Balboa, Ancon, Republic of Panama, 3 School of Fisheries, Aquaculture and Aquatic Sciences, Auburn University, Auburn, AL, United States of America, 4 Red Hills Fishery, Boston, GA, United States of America, 5 National Institute of Aquatic Resources, Technical University of Denmark, Kgs. Lyngby, Denmark

* mab0205@auburn.edu

**Data Availability Statement:** Sequence data is available on GeneBank BioProject PRJNA1100459. Computational scripts used for the analyses are

## Abstract

Temperature is one of the most relevant factors influencing the development of aquatic species, making it a key parameter to consider for aquaculture. Largemouth bass (LMB; *Micropterus spp.*) are highly relevant for human consumption and sport fishing, representing one of North America's most important freshwater fisheries. Yet, questions remain on how LMB raised in recirculating aquaculture systems (RAS) respond to different temperatures. The main objective of this study was to determine the impact of thermal rearing conditions (21˚C, 24˚C, and 27˚C) on gene expression of Florida and Northern LMB larvae at 8- and 28-days post hatch (DPH). Using *de novo* transcriptomes as a reference, our results suggest that gene expression differences for Florida LMB were mostly associated with temperature, while differences for Northern LMB were controlled by temperature and developmental stage. In general, both lineages showed activation of molecular pathways associated growth, such as development of muscle, nervous system, and vascular system. There were molecular signatures of stress with warming as well, including immune function, apoptosis, regulation of inflammation, and heat shock proteins. Florida LMB showed large differences between temperatures at both stages, while differences were much larger for Northern LMB at 28 DPH, specifically for individuals reared at 27˚C. The results from this study are in line with previous phenotypic studies that indicated faster growth at warmer temperatures and better performance of Northern LMB raised in RAS. Overall, this study exemplifies how controlling developmental temperatures during the critical early life stages can be essential to guarantee the success of commercial hatchery production techniques.

available at: https://github.com/evofish/
Largemouthbass-transcriptomics. Complete tables
of differentially expressed genes and gene ontology
analyses are available at: https://doi.org/10.6084/
m9.figshare.25620786.v1 (Supplementary Data 1
& 2).

**Funding:** This project was supported by funds
from Red Hill Farms. Additional support provided
by Agriculture and Food Research Initiative
Competitive Grant from the USDA National Institute
of Food and Agriculture hatch projects 1013854 to
IAEB, ALA-016-1-19075 to AMK, and ALA016-1-
19053 to LAR. The funders had no role in study
design, decision to publish, or preparation of the
manuscript.

**Competing interests:** The authors have declared
that no competing interests exist.

## 1. Introduction

Fishes largely depend on the conditions of their surrounding environment for development and reproduction, and temperature is considered one of the key factors influencing sensitive life stages [1, 2]. In this regard, thermal conditions are fundamental for biochemical processes, and since most fishes are ectotherms, they have a direct influence on the rate of metabolic activity of a particular individual [3, 4]. In fish offspring, temperature drives traits such as time to hatch, developmental rates, and growth patterns, while conditions beyond thermal optima can lead to increased presence of deformities in larval stages [5–7]. In addition, studies have focused on how warming affects swimming performance [8] and immunity [9] during fish larval stages, which can have direct consequences for survival and predator/prey dynamics.

Taking this into account, temperature is one of the essential parameters to consider regarding husbandry and rearing of aquatic species of commercial interest. Yet, in many instances, commercially relevant species are reared in semi-natural conditions, where abiotic and biotic parameters can show high variability, especially as climate change is projected to promote acute thermal fluctuations in coming decades [10]. Further, several species of freshwater fish are reared in earthen ponds during early life stages, making them vulnerable to changing abiotic conditions as well as predation, which can have detrimental effects on production efficiency [11–13]. Thus, controlling the effects of temperature and avoiding predation through recirculating aquaculture system (RAS) technology during early life stages could lead to improved production of freshwater species of interest.

Largemouth bass (LMB) *Micropterus spp.* is the most widely distributed black bass [14, 15] and one of the most popular sport fishes cultivated in the United States for pond stocking since the 1890s [16, 17]. In fact, the economic impact of LMB pond stocking reached over $14.45 million in 2013 by 176 farms and climbed to $27.46 million in 2018 by 195 farms [18]. In addition to sport fishing, their use as a food fish for the aquaculture industry is also increasing (0.2 thousand tonnes live weight in 2002 to 621.3 thousand tonnes of live weight in 2020), making up 1.3% of the total percentage of finfish in inland aquaculture worldwide [19]. However, despite the increasing demand for LMB, larval fish are still reared in earthen ponds, with significant mortality due to predation and temperature changes. Given the high economic relevance of this species, it is essential to identify optimal conditions, specifically thermal ranges, for rearing early life stages of this species.

Due to its commercial importance, previous studies have addressed patterns of gene expression in relation to LMB reproduction [20], exposure to pollutants [21, 22], consumption of different diets [23, 24], and immune responses during pathogen exposure [25]. Fewer studies have focused on the effects of temperature on early development. This requires careful analyses given that the variable developmental conditions experienced early in life can lead to long-term effects of thermal tolerance and temperature preference [26]. Further, members of the LMB complex have been introduced to different locations globally, and the diverse genetic lineages of LMB available for commercial husbandry have not been specified in previous studies. For instance, previous studies on juveniles and adults (i.e., individuals >80g) suggest that LMB acclimated to 26°C prefer temperatures between 26 and 29°C, but that they can survive temperatures of up to 35°C [27, 28]. Maximum thermal tolerance of LMB indicates that there are slight differences among the Florida (39°C) and Northern lineages [37°C; 29]. Despite this apparent high tolerance, individuals fail to gain weight efficiently and decrease their food consumption at temperatures >30°C in earthen ponds [27]. Moreover, a recent study on Florida and Northern LMB larval performance at three different temperatures (21°C, 24°C, and 27°C) for individuals acclimated to 20 to 22°C, found that the rate of yolk consumption was faster with warming and growth was highest at 27°C [30]. In this comparison, Northern LMB larvae

had higher survival than the Florida lineage during early development at the warmest temperatures [30]. In contrast, a comparison between both groups at 33˚C showed lower thermal stress for Florida samples under acute (3 days) and chronic warming (2 months), based on 13 biomarkers, including cortisol and heat-shock proteins [31]. One of the few studies that has evaluated genome-wide gene expression of Northern LMB juveniles in relation to different temperatures found that individuals exposed to warm conditions showed activated pathways related with antioxidants, inflammation, apoptosis or energy metabolism, and compromised immune responses when facing pathogens [32]. Despite these advances, questions remain on how temperature influences underlying molecular processes in Florida and Northern LMB during the "critical" early life stages (i.e. <30DPH) when fishes undergo drastic changes in food sources, morphology, and elevated mortality [33].

Given that aquaculture production will increase to a projected 100 million metric tons by 2030 and improve employment opportunities in developing countries [19], it is essential to broaden our understanding of how thermal conditions influence developmental stages of fishes. As such, this study aimed to evaluate differential gene expression (via TagSeq) of two lineages of *Micropterus* (Florida and Northern LMB) across different rearing temperatures based on samples originating from the experiment presented in Aguilar et al. [30]. Previous studies have reported significant genetic differences between these two groups [34, 35], but in many cases, reports do not specify the genetic lineage that is being evaluated. Recently, they have been split into two species based on genomic approaches: Florida LMB, now recognized as *Micropterus salmoides*, and Northern LMB, recognized as *Micropterus nigricans* [35]. Based on this change, Florida LMB is found from North Carolina to the Florida Peninsula, to the Apalachicola River in the West. Northern LMB has a broader distribution, from the Laurentian Great Lakes to Northern Mexico and the Mississippi drainage to the East [35]. There are also slight morphological differences that characterize the two species. Florida LMB tend to be slightly larger and have 27–34 scales around the peduncle and 65–77 lateral scales, while Northern LMB have 24–32 around the peduncle and 58–69 lateral scales [36]. They still have the capacity to hybridize in areas of overlap as well as in captivity [29, 34, 35]. However, for the purpose of this study, they will be referred to as different "lineages" since the taxonomic status of the group is beyond the scope of the research questions, and producers might not be familiar with the change in nomenclature. Specific questions of this study include: i) What are the differences in gene expression among different temperature conditions (21˚C, 24˚C, and 27˚C)? ii) What are the differences between larvae at 8- and 28-days post hatch (DPH)? iii) Are there differences in thermal tolerance between Florida and Northern LMB as measured by differential gene expression? Considering that fish at higher latitudes tend to develop faster as they have shorter growing seasons [i.e., counter gradient growth hypothesis; 37], we hypothesize that Northern LMB will have faster growth than Florida LMB, and that this will be reflected in the activation of molecular pathways associated with development. By expanding the genomic resources for two lineages of *Micropterus spp*., this study represents an essential contribution to improving LMB aquaculture worldwide.

## 2. Materials and methods

### 2.1. Experimental design

Fish experimentation protocols were reviewed and approved by the Institutional Animal Care and Use Committee (IACUC# 2020–3772) at Auburn University. An experimental overview is illustrated in Fig 1. Florida and Northern LMB used in the present study were raised at Red Hills Fishery in Boston, Georgia, USA (30.8478˚N, -83.7606˚W).

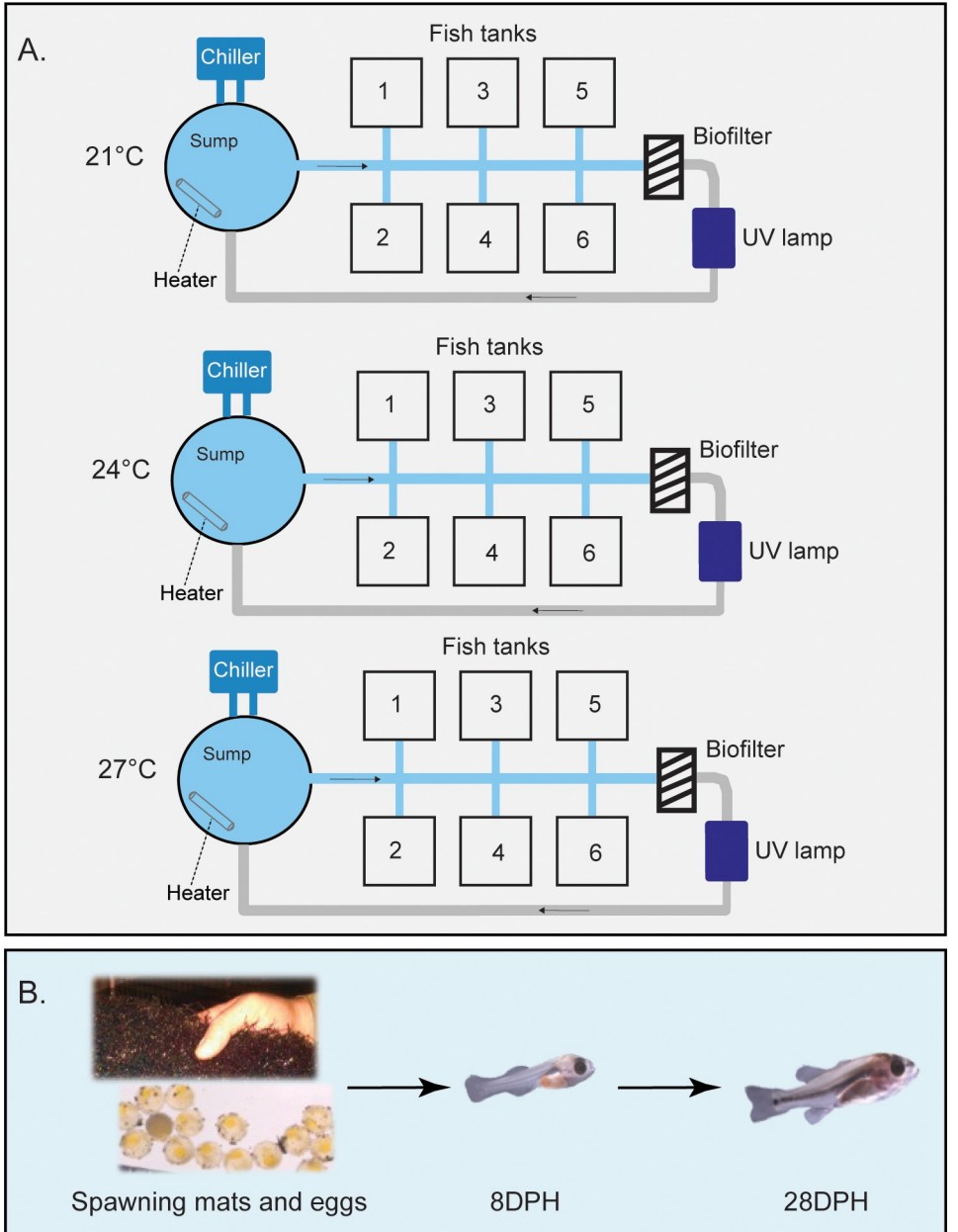

**Fig 1. Experimental overview.** Largemouth bass (LMB) broodstock fish were reared at a commercial farm, and eggs were transported to Auburn University. LMB offspring were reared using recirculating aquaculture systems (RAS) at different temperatures: Control (21˚C), Intermediate (24˚C), or Warm (27˚C; A). Samples were collected on 8- and 28-days post hatch (DPH) for molecular analyses (B). Photographs by co-authors JS and IAEB.

Adult Florida LMB were fed goldfish, while Northern LMB were fed a combination of goldfish and a broodstock diet (Richloam Bass Fry #14: 64% fish meal, 13% corn gluten meal, 2% fish oils, 1% vitamin supplements) at ~2% body weight per day. Each lineage was raised in an indoor facility in concrete raceways of 27 × 3 × 1 m with a flow rate of 90 gal/min. Fish were reared on photoperiodic cycles of 10 h light and 14 h dark for the first four weeks, then 8 h light for three to four weeks, then back to 10 h light for two weeks, and finally 14 h light for the last two weeks to initiate spawning. Spawntex spawning mats (Pentair Aquatic Eco-Systems,

Apopka, FL, USA) were evenly distributed along the entire length of the raceway. Florida broodstock comprised 34 males and 40 females (length: 321 to 584 mm; weight: 0.48 to 3.97 kg), while Northern broodstock comprised 74 males and 56 females (length: 330 to 481 mm; weight: 0.56 to 1.99 kg). During the artificial spawning season (1 Sept 2020 to 27 Oct 2020), water temperature and dissolved oxygen (DO) ranged from 19.5 to 23.0˚C and 8.15 to 10.23 mg/L, respectively for Florida LMB and 19.4 to 22.7˚C and 10.17 to 12.25 mg/L, respectively for Northern LMB. At the time of embryo collection, mean nitrite was 0.019 mg/L, mean nitrate was 0.8 mg/L, and mean total ammonia nitrogen (TAN) was 0.07 mg/L.

Spawning mats were checked twice daily, and once eggs/embryos were detected, they were transported to the Auburn University E.W. Shell Fisheries Center (32.6526˚N, -85.4859˚W) in 114 L coolers with ~40 L of water from the raceways. Six spawning mats containing Florida LMB eggs/embryos were transported on September 25, 2020, and ten spawning mats containing Northern LMB eggs/embryos were transported on October 5, 2020. Upon arrival, the spawning mats containing eggs/embryos were held at 21.5 ± 0.5˚C in 75 L black aquaria, resting 15 cm below the surface. Once ~50% hatch occurred, larvae were gently pipetted into 30 mesh baskets (25 × 33 × 13 cm; i.e., 15 baskets for each lineage). Each basket was stocked with 2,200 yolk-sac larvae that equally represented the different spawning mats; ~367 larvae per spawning mat for Florida LMB and ~220 larvae for Northern LMB were connected to a single recirculating aquaculture system (RAS) at 21˚C. Once all baskets were stocked, 20 baskets (10 Florida LMB, 10 Northern LMB) were transferred to two separate RAS (i.e., at 21˚C; n = 5 per lineage per system). The aquaria in each of these RAS were slowly acclimated to their desired temperature (24 or 27˚C) by adjusting the flow rate to 0.5 L/min (~1˚C/h). The other 10 baskets (5 Florida LMB, 5 Northern LMB) remained in the 21˚C RAS. Larvae were released from the baskets once >50% reached the swim-up stage, and the aquarium water level was reduced to 17 L.

Three temperature treatments (Control = 21˚C, Intermediate = 24˚C, or Warm = 27˚C) were chosen based on the temperatures experienced by the spawning adults (19˚C-23˚C) and the temperatures experienced during the egg incubation (21˚C). Each one of the thermal treatments were setup in one RAS (i.e., three separate RAS total), and the larvae used for the gene expression analyses come from four to six different tanks within each recirculating system (Table 1). The RAS were equipped with a UV sterilizer, bead and bag filters, heaters, and chillers for temperature control [30]. Temperature and DO were checked twice daily with a YSI multiparameter meter (YSI, Yellow Springs, OH, USA), and the temperature fluctuation was minimal (± 0.2˚C). Flow rate of water was set to ~7 L/min. Rearing of offspring took place under a 12 h light/12 h dark photoperiod at ~250 lux, pH ranged from 7.2 to 7.7, alkalinity was between 95 and 125 mg/L $CaCO_3$, and nitrite and nitrate levels between 0 to 0.02 mg/L.

Larvae were fed before the yolk was completely absorbed (~3 DPH), to get them stimulated to eat and become more efficient at foraging. Larvae were fed premium Grade A *Artemia*

**Table 1. Number of samples (i.e., individual larvae) from Florida and Northern Largemouth Bass (LMB) that were sequenced for each treatment, and the number of different tanks that the samples were distributed in to avoid pseudo-replication.** Differences in sample sizes per treatment were associated with the quality of RNA extractions, as only samples with an RNA integrity > 7 were included for library preparation.

| Temperature | 8DPH | | | | 28DPH | | | |
|---|---|---|---|---|---|---|---|---|
| | Florida | | Northern | | Florida | | Northern | |
| | n | Tanks | n | Tanks | n | Tanks | n | Tanks |
| 21˚C | 8 | 5 | 6 | 4 | 7 | 4 | 7 | 4 |
| 24˚C | 8 | 5 | 10 | 5 | 9 | 5 | 10 | 6 |
| 27˚C | 8 | 5 | 8 | 5 | 8 | 5 | 7 | 4 |

(Brine Shrimp Direct, Ogden, UT, USA) at a concentration of 2 *Artemia*/mL every 2h from 6 am to midnight. Mortalities were removed daily along with excess feed and fecal matter.

## 2.2. Transcriptome sequencing, assembly, and annotation

One adult individual per lineage was collected from Red Hills Fishery in Boston, GA. Individuals were euthanized by cervical transection and pithing, and seven tissues were extracted and preserved in RNA/DNA Shield (Zymo, Irvine, CA, USA): liver, brain, gill, muscle, fins, eye, and heart. These tissues were used to ensure the reference transcriptome contained a representative number of genes. Each tissue was individually extracted using the Quick-RNA tissue kit (Zymo, Irvine, CA, USA), following the manufacturer's instructions. The extracted RNA from the different tissues was consolidated into a single "master" mix, in the following proportion of volume, based on the concentration of the individual RNA extractions: 30% brain, 20% liver, 10% muscle, 10% fins, 10% eye, 10% heart, and 10% gill. The RNA mix was sent for sequencing and library preparation to Novogene (Sacramento, California, USA). Library preparation was done using the Illumina TruSeq RNA Kit, while the sequencing of 150 bp paired-end reads was done with the Illumina NextSeq 5000. Raw reads for the transcriptome are available at NCBI BioProject PRJNA1100459 (SUB14375617).

Raw reads were filtered for quality and sequencing, and Illumina adapters were removed using Trimmomatic [38]. Each transcriptome was assembled separately for the corresponding lineages using Trinity [39]. The parameters were set to default, except for the minimum contig length, which was set to 300 bp. After contigs were obtained, the program CD-HIT [40] was used to cluster reads by similarity (95%) and remove redundancy among contigs. Annotation of the transcriptomes to gene level and Gene Ontology category was done with BlastX using the UniProt database as a reference [accessed December 2021; 41]. The contigs that failed to match with the UniProt database were compared to the annotated cDNA of zebrafish, *Danio rerio* (ENSMBL GRCz11) and blue tilapia, *Oreochromis aureus* (ENSMBL ASM587006v1; Accessed January 2022) using BlastN [41]. For all Blast searches, matching of reads to the reference had a threshold e-value of $<e\text{-}10$, and a maximum of five matching sequences were retained. The highest match was chosen based on the percent identity, the bit-score, and the percent of sequence coverage between reference and query. The program BUSCO [42] was used to evaluate the completeness of the sequenced transcriptomes, using the reference library for bony fishes (Actinopterygii) provided by the program.

## 2.3. Larval sampling

Individual larvae were sampled at two developmental time points, 8 and 28 DPH, and euthanized with 150 ppm MS-222 (tricaine methanesulphonate; Argent Laboratories Inc., Redmond, WA, USA). These timepoints were selected based on the time when LMB larvae have a bout of mortality (~8 DPH) and by the time larvae are typically weaned unto commercial diet (~28 DPH). The individual larvae were randomly selected from the individual aquaria, and between 6 and 10 individuals were collected for each treatment (Table 1). A total of 96 samples were collected for the analyses of differential expression, and these were initially snap frozen in liquid nitrogen and then transferred to -80°C until further processing.

Individual larvae were extracted using the Quick-RNA tissue kit (Zymo, Irvine, CA, USA), following the manufacturer's protocols. The only modifications of the protocol were the time to incubation of samples with Proteinase K (overnight at 32°C) and the adjustment of volume for the final elution to 60 μL. Individual larvae extractions that had a concentration higher than 100 ng/μL of RNA, and an integrity (RIN) of 6.5 or above were used for library preparation. The library preparation and sequencing were completed for individual larvae, to avoid

any biases from pooling RNA from different samples. TagSeq library preparation and sequencing were conducted at the GSAFU of the University of Texas in Austin (USA), following the protocol delineated by Meyer et al. [43, 44]. Previous studies indicate that Tag-Seq is a very reliable method for assessing differential expression, showing strong correlation with control mRNA genes [43, 45]. Sequencing was performed with an Illumina Hiseq4000, and fragments were 50 bp single end. Raw reads for the gene expression reads are available through NCBI BioProject PRJNA1100459 (SUB14448778).

## 2.4. Analysis of gene expression

The resulting sequences were analyzed following the Tag-Seq bio-informatic pipeline (scripts and documentation available at: https://github.com/z0on/tag-based_RNAseq). Sequences from a single individual were received as two different sequence files and were concatenated into a single Fasta file using the "cat" function in Unix. Adaptor trimming of each sample was performed with the script *tagseq_clipper.pl* and removal of low-quality reads was performed via the program Cutadapt (minimum length -m 25bp and quality cutoff -q 15; Martin, 2011). Samples of each lineage were mapped to their respective transcriptome using Bowtie2 [46]. Summarization of Tag-Seq read counts was accomplished using the script *samcount_launch_bt2.pl*, where the reads were collapsed into "isogroups" generated during the Trinity transcriptome assembly. The final read counts contained in the SAM files were summarized into a single table using the *script expression_compiler.pl*, where a table of reads by isogroup for each gene in the transcriptome was exported as a text file for all the subsequent analyses.

Differential gene expression analyses were conducted with the DeSeq2 platform [47] in RStudio (v 3.4.4; R Core Team). Since differences in sequencing effort can cause outliers in the number of Tag-Seq reads, a heatmap of sample distance based on raw read counts was elaborated. Any individual outlier in the distance matrix was eliminated to prevent any differences not caused by the experimental treatments (no samples were detected as outliers based on read number). For gene expression analyses, a likelihood ratio test (LRT) was conducted to evaluate the relative contribution of the different experimental factors to the observed gene expression: temperature (21°C, 24°C or 27°C), age (8 or 28 DPH), and experimental aquarium tanks. First, the effect of all conditions together was analyzed using the LRT. Second, the relative contribution of one of the variables was estimated, controlling for the other two variables (e.g., the effect of temperature, excluding the effect of age and experimental tanks). The RNAseq reads were normalized following the standard DESeq2 method of median-of-ratios, where the main factor being considered is the sequencing depth for specific genes compared across all the different individual conditions (thermal treatments and DPH) for each lineage. In addition to the LRT, analyses of pairwise differential expression were performed, where each of the temperature treatments was compared to one another using the "Contrast" function in DESeq2. The "Contrast" function is used to evaluate if differences in log2fold change between two groups are statistically different from zero for each gene. For the LRT and the pairwise contrast analyses, samples were significant if they had an adjusted p-value of <0.01 after Benjamini-Hochberg correction. It is relevant to highlight that all the comparisons were limited to samples of the same lineage, as there were differences in the diets between Florida and Northern LMB throughout development that could bias any interpretation of direct comparisons between the groups.

The analyses of pairwise contrasts were explored further through a test of Gene Ontology (GO) Enrichment. For this, a Mann-Whitney U test (MWU) was used to determine if some activated or deactivated gene categories were disproportionately represented in the pairwise contrasts, using the R package GO-MWU [48]. The pipeline for the analysis and the associated

scripts are available at https://github.com/z0on/GO_MWU. The input values for the analyses were the raw "log2fold change" scores obtained from the pairwise comparisons, as recommended by the user guideline. Using the raw log2fold change, the GO-MWU program allowed estimating if there was a disproportionate representation of categories at the top (activated processes) or bottom (deactivated processes) of the measurements. The tests were conducted for each pairwise comparison, and categories were significantly enriched if they had a False Discovery Rate (FDR) <1%. The corresponding GO categories for each one of the isogroups were obtained from the annotation of the transcriptome to the UniProt dataset, and the zebrafish and Nile tilapia genomes (as described above). Each test was completed separately for each of the different GO subcategories: Biological Process (BP), Cellular Component (CC), and Molecular Function (MF).

Weighted Gene Network Correlation Analyses (WGCNA) were conducted for both lineages. The WGCNA clusters genes that are being expressed in a similar way into modules to find broader functional patterns influenced by the experimental treatments [i.e., temperature and developmental stage; 49]. The networks were signed, which means modules were composed of genes that were being expressed in the same direction (i.e., upregulated or downregulated for specific treatments). The modules had a minimum of 50 genes and a maximum of 5,000. The resulting modules were then correlated to the specific traits: the three different temperatures and the two life stages.

Computational scripts used for the analyses of transcriptome assembly, differential gene expression, and WGCNA are available at: https://github.com/evofish/Largemouthbass-transcriptomics.

## 3. Results

### 3.1. Transcriptome assembly

Both transcriptomes were comparable in terms of number of transcripts (Florida = 118,765; Northern = 123,744) and size distribution of fragments (Fig 2). Clean reads were assembled with Trinity and clustered by similarity using CD-HIT, resulting in a reference transcriptome of Florida LMB, with a maximum contig length of 26,875 bp, average length of 1,226.80 bp, and N50 of 2,408 bp (Table 2). Meanwhile, the transcriptome of Northern LMB had a maximum contig length of 25,593 bp, average length of 1,187.40 bp, and N50 of 2,308 bp (Table 2). Based on the reference database for bony fishes from BUSCO, the Florida LMB transcriptome had a completeness of 89.5%, while Northern LMB had 90%. Using the UniProt database and the ENSMBL genomes as a reference, the annotation of the Florida LMB transcriptome resulted in successful matches for 39,706 of the contigs (33.43%). In comparison, Northern LMB had 40,407 annotated contigs (32.65%). Results of the transcriptome assembly and annotation are summarized in Table 2 and show how the number and length of contigs is comparable among the two *de novo* transcriptomes. Differences among the two *denovo* transcriptomes based on the number of contigs, or the number of complete or fragmented BUSCOs are most likely associated with stochastic factors such as the effectiveness of sequencing and assembly. The two species are closely related and still capable of hybridizing [35], thus we do not expect that these differences are associated with differences in chromosomal rearrangements or genome duplications.

### 3.2. Analyses of differential gene expression

For differential expression analyses across different temperatures and ages, the mean number of reads per sample was 1,142,347, with a minimum of 527,797 and a maximum of 2,921,704. The sample-to-sample heatmap with the raw read counts suggested no outliers in the data

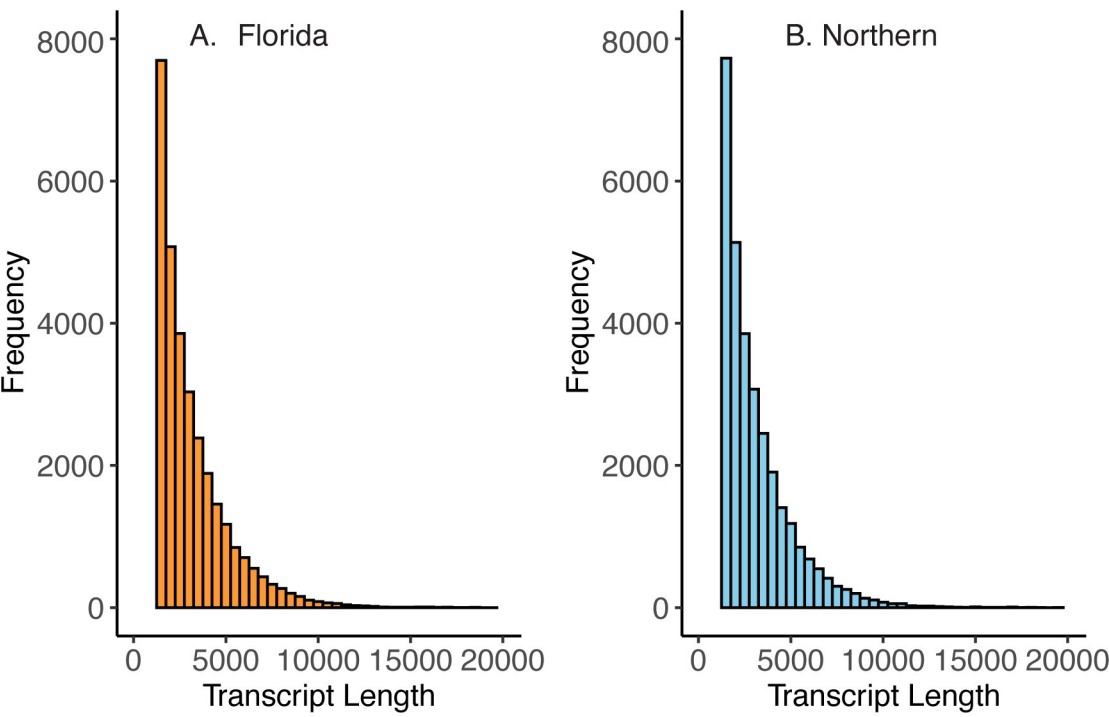

**Fig 2.** Histograms showing the length and frequency of the assembled contigs for Florida (A) and Northern (B) Largemouth Bass (LMB) transcriptomes.

based on the number of reads. Thus, all samples sequenced were included in the analyses of expression, GO enrichment, and WGCNA.

Expression analyses with the LRT allowed the effects of temperature, developmental stage (DPH), and rearing tanks to be disentangled. For Florida LMB, comparisons among the different temperatures, excluding the effect of stage and tank, resulted in 7,651 differentially expressed genes (DEG), suggesting this was the dominant factor promoting differences in expression. This was followed by the effect of stage, which resulted in 488 DEGs, excluding the effects of temperature and rearing tank. Finally, the rearing tank had a much smaller effect, with only 34 DEGs when excluding temperature and developmental stage. This is consistent with the PCA of the variance stabilized data, where the largest axis of differentiation PC1 (59% of variance) showed the largest differences between 21˚C and 27˚C (Fig 3A). For Northern LMB, the largest effect was associated with stage with 6,434 DEGs, followed by 5,010 DEGs

**Table 2. Number of contigs, average contig length, N50 and BUSCO results of completeness for the reference transcriptomes of Florida and Northern Largemouth Bass (LMB).**

|  | FLORIDA | NORTHERN |
|---|---|---|
| **TRINITY CONTIGS** | 165,215 | 172,956 |
| **CD-HIT CONTIGS** | 118,765 | 123,744 |
| **AVERAGE LENGTH** | 1,226.8 | 1,187.4 |
| **MAX LENGTH** | 26,875 | 25,593 |
| **N50** | 2,408 | 2,308 |
| **COMPLETE BUSCOS** | 3259 (89.5%) | 3279 (90.0%) |
| **FRAGMENTED BUSCOS** | 113 (3.1%) | 83 (2.3%) |
| **MISSING BUSCOS** | 268 (7.4%) | 278 (7.7%) |

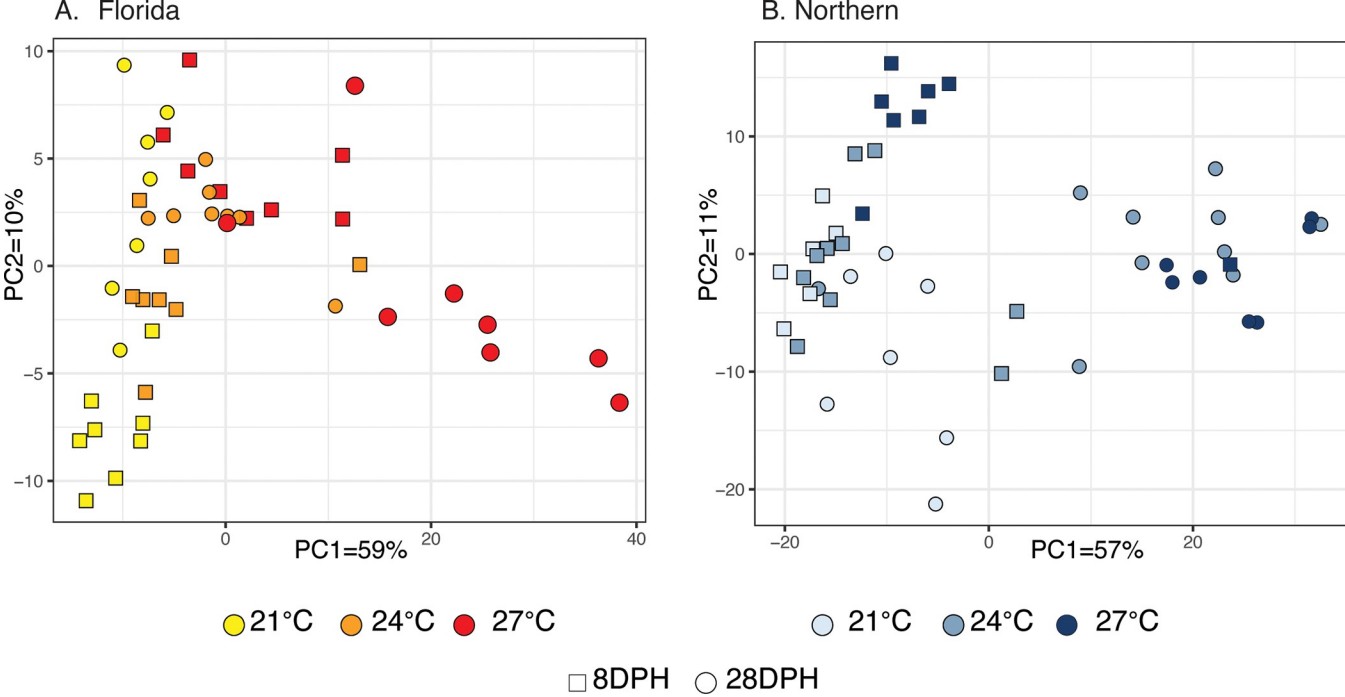

**Fig 3. Principal Component Analysis (PCA) for all transcripts, based on the DESeq2 analysis.** The PCA shows differences among temperature treatments (21, 24, or 27˚C) and developmental stage (8- or 28-days post hatch-DPH) for Florida (A) and Northern (B) Largemouth Bass (LMB), based on gene expression of all transcripts.

associated with temperature, and only 20 DEGs associated with rearing tank effects. This again coincides with the results for the PCA of the variance stabilized data, which showed PC1 (57% of variance) was mostly influenced by developmental stage and temperature, as the largest differences were observed between Control and Warm conditions (24˚C and 27˚C) at 28 DPH (Fig 3B). As such, the LRT and PCA results confirm that tank effects were negligible and that the experimental conditions of temperature and stage were the main drivers of differentiation in gene expression during larval development for both lineages.

The number of DEGs resulting from pairwise contrasts and the corresponding GO enrichment results are summarized in Table 3. The unique and overlapping DEGs for each comparison are presented in Fig 4. The genes that were significantly different for all comparisons are available in Supplementary Data 1, and the specific GO categories in Supplementary Data 2. These comparisons show that the largest number of DEGs were observed for Florida and Northern LMB larvae at 28 DPH. The patterns for Florida LMB, at 8 DPH, indicated a large difference between the lowest temperature (21˚C) and the two warmer temperatures (1,927 DEGs at 24˚C and 2,543 at 27˚C). Individual genes that were activated for the warm temperatures were associated with apoptosis and tumor suppression (*aatka*), signal transduction (*camk2*, *gbb1*, *opn1*), ion transport (*atp1b3a*, *frrs1*, *tmem63a*), development and growth (*dkk3a*, *fgf14*, *igf2bp1*, *kmt2a*), eye development (*gngt1*), and neuronal development (*geph*, *chata*, *omga*, *pcloa*, *nefl*, *syt2a*). Overlapping GO categories that were activated in both 24˚C and 27˚C were associated with neuro-transmission (e.g. *Glutamate receptor activity*, *Neurotransmitter secretion*, *Neurotransmitter transport*, *Presynapse*, *Presynaptic active zone*, *Synapse*, *Synaptic membrane*), cell-signaling (*Cell-cell signaling*, *G protein-coupled receptor signaling pathway*, *G protein-coupled receptor activity*, *Signal release*, *Signaling*, *Signaling receptor activity*), ion transport (*Potassium transport*, *Sodium transport*, *Sodium channel complex*, *Calcium*

**Table 3. Differentially expressed genes (*p*<0.01) and several Gene Ontology (GO; FDR<1%) for two lineages of Largemouth Bass (LMB).** Results for Florida (below diagonal) and Northern (above diagonal and in bold) LMB larvae across different temperatures and corresponding GO enrichment results in italics (GO Categories: BP–Biological Process, CC- Cellular Component, MF- Molecular Function), at 8 (A) and 28 (B) days post hatch (DPH).

**A.**

| 8DPH | 21°C | 24°C | 27°C |
|---|---|---|---|
| **21°C** | - | **5** <br> *4 BP, 4 CC, 3 MF* | **240** <br> *56 BP, 14 CC, 32 MF* |
| **24°C** | 1,927 <br> *80 BP, 30 CC, 45 MF* | - | **441** <br> *43 BP, 8 CC, 22 MF* |
| **27°C** | 2,543 <br> *41 BP, 28 CC, 31 MF* | 121 <br> *1 BP, 7 CC, 1 MF* | - |

**B.**

| 28DPH | 21°C | 24°C | 27°C |
|---|---|---|---|
| **21°C** | - | **2,528** <br> *125 BP, 36 CC, 55 MF* | **6,971** <br> *209 BP, 39 CC, 71 MF* |
| **24°C** | 158 <br> *21 BP, 6 CC, 5 MF* | - | **18** <br> *27 BP, 12 CC, 21 MF* |
| **27°C** | 7,831 <br> *22 BP, 4 CC, 29 MF* | 3,799 <br> *7 BP, 4 CC, 12 MF* | - |

ion transport, *Calcium channel complex*, *Channel activity*), and cellular structure (*Cytoskeleton*, *Microtubule based movement*, *Microtubule motor activity*). Meanwhile, the contrast between 24°C and 27°C resulted in 121 DEGs, where the GO enrichment analysis showed the activation of categories related with neurotransmission (*Perikaryon*, *Synapse*) at 24°C and DNA replication (*DNA integration*, *Nuclear chromosome*) at 27°C.

The patterns of DEGs for Florida LMB were different at 28 DPH (Table 3B), as the comparison between 21°C and 24°C resulted in a modest number of DEGs (158), where genes such as cellular structure (*tbcela*, *tmod4*), cellular respiratory chain (*cox7a1*), muscle development (*mylpf*, *atp2b2*, *pvalb4*), glucogenesis (*aldoa*), and development of nervous system (*bcan*) were activated at the warmer temperature. GO categories that were activated at 24°C were associated with muscle formation (*Skeletal myofibril assembly*, *Actin filament binding*), processing of nucleic acids (*Chromosome*, *DNA metabolic process*, *Nuclear chromosome*), and immunity (*Immune system process*), but also de-activation of energy regulation (*Regulation of ATP metabolic process*) was observed. In contrast, the comparison between 21°C and 27°C resulted in 7,831 DEGS, the largest number for any comparison in the study. Genes activated at 27°C at 28 DPH included functions such as regulation of transcription (*kansl3*, *lorf2*, *asmt2*, *prdm15*), growth and development (*dhx30*, *etc2*, *hpn*, *poc1a*, *shox2*), cell division and replication (*cdc45*, *ska3*, *pole*), development of nervous system (*camk2b*, *lhx6*, *neto1*, *tbc23*, *sox7*), bone growth (*bglap*, *fos*), immunity and inflammation (*ifih1*, *pgdh*), and energy regulation (*brsk2b*, *itln*). GO enrichment categories at 27°C at 28 DPH were associated with formation of lipid layers (*FFAT motif binding*), DNA replication (*DNA replication initiation*, *DNA recombination*, *DNA integration*, *DNA polymerase activity*), cell division (*Mitotic cell cycle*, *Cell Cycle*), and development of nervous system (*Axonal fasciculation*). Interestingly, the warmest temperature also showed deactivation of neurotransmission (*GABA receptor activity*, *Synapse*, *Synapse assembly*, *Synapse organization*, *Nerexin family protein binding*, *Glutamate receptor signaling pathway*) and transmembrane transport (*Exocytosis*, *Organic acid transmembrane transporter activity*, *Voltage-gated calcium channel activity*).

In the case of Northern LMB, the pairwise contrasts suggested a much more modest number of DEGs for comparisons at 8 DPH. The comparison between 21°C and 24°C resulted in 5

## A. 8 DPH -FLO

### 21°C vs. 24°C

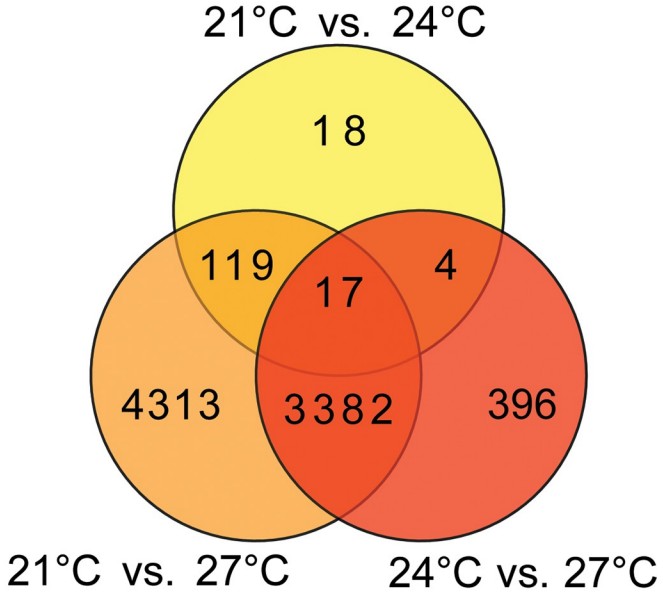

## B. 28 DPH - FLO

### 21°C vs. 24°C

21°C vs. 27°C    24°C vs. 27°C

## C. 8 DPH - NOR

### 21°C vs. 24°C

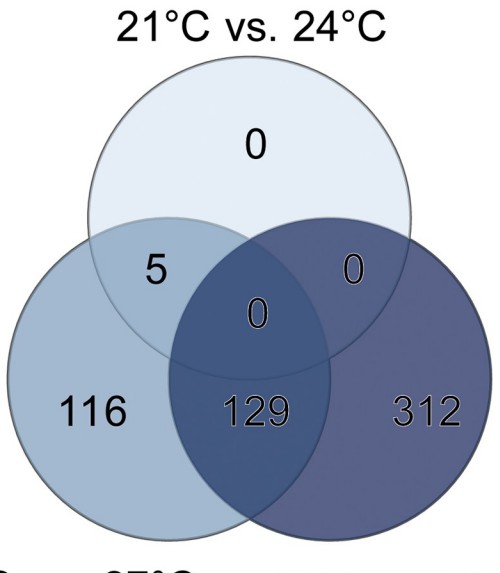

21°C vs. 27°C    24°C vs. 27°C

## D. 28 DPH - NOR

### 21°C vs. 24°C

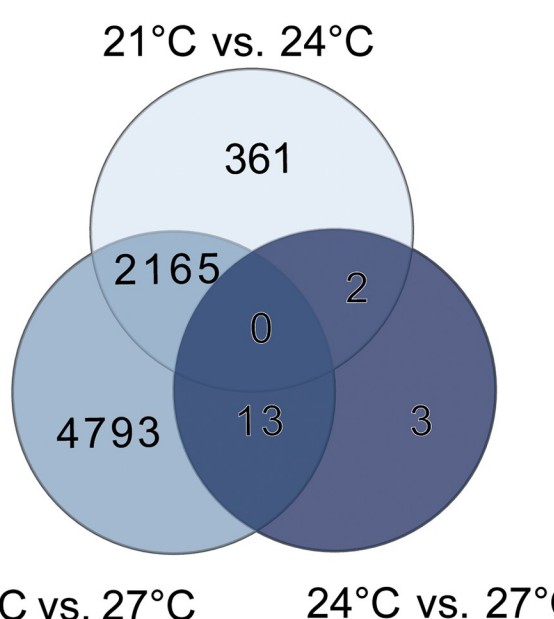

21°C vs. 27°C    24°C vs. 27°C

**Fig 4. Venn diagrams showing the number of differentially expressed genes.** The comparisons were among different temperature treatments (21˚C, 24˚C, and 27˚C) at different developmental stages (8- and 28-days post hatch- DPH) for Florida (A- 8DPH; B- 28DPH) and Northern (C- 8DPH; D 28DPH) Largemouth Bass (LMB) larvae.

DEGs, including the activation of immunity (*cd59*, *tnip1*) and bone development (*ucma*) genes. Meanwhile, GO categories activated in the warm temperature included oxygen transport (*Oxygen transport*, *Hemoglobin complex*) and protein synthesis (*Biosynthetic process*, *Organonitrogen compound biosynthesis*). Meanwhile, the comparison between 21˚C and 27˚C resulted in 240 DEGs, while the comparison between 24˚C and 27˚C showed 441 DEGs. Genes that were activated at the warmest temperature (27˚C) were associated with cell death and tumor suppression (*clu*, *soul3*), cell division (*sinhcaf*), immunity (*ef1a1*), protein destabilization (*serf2*), anti-inflammatory response (*tsc22d3*), and neurological development (*vsnl1*). The GO categories activated at 27˚C included oxygen transport (*Oxygen transport*, *Hemoglobin complex*), growth and development (*Organ growth*), immune response (*Positive regulation of B cell activation*, *Peptidoglycan muralytic activity*, *Regulation of defense response*), muscle development and contraction (*Actomyosin structure organization*, *Muscle Cell Development*, *Muscle tissue morphogenesis*, *Myosin II complex*, *Actin filament binding*, *Sarcomere*, *Straited muscle contraction*, *Structural constituent of muscle*), eye development (*Structural constituent of eye lens*), and cardiovascular development (*Cardiac cell development*, *Cardiac muscle contraction*, *Ventricular system development*).

Meanwhile, the comparisons among Northern LMB larvae at 28 DPH resulted in a much higher number of DEGs, with 2,528 between 21˚C and 24˚C and 6,971 between 21 and 27˚C, with a large level of overlap among them (Fig 4D). Categories that were activated in the warm conditions (24˚C and 27˚C) with respect to 21˚C were associated with cellular respiration and electron transport chain *(cox4i2*, *cox5a*, *cox6b*, *acat1*, *ndufa1)*, muscle activity and development *(acta1*, *adss1*, *lmna*, *mybpc3b*, *myom1b*, *myom2a*, *myoz1)*, cardiovascular development *(actr2a*, *angptl1b*, *hopx*, *myhb*, *myl1*, *myl2*, *myl3*, *myl7*, *pim1)*, lipid metabolism and organization *(aebp1*, *hdlbpa)*, apoptosis and inflammation *(aimp1b*, *hypk*, *mcl1b*, *rassf1)*, cell growth and differentiation *(gxylt2m*, *id1*, *igfbp2a*, *igfbp7*, *mmp2*, *notch2*, *notch3)*, nervous system development and activity *(itpkcb*, *ipo9*, *kctd12.2*, *rtn2)*, energy metabolism *(ampd3b*, *pygma)*, immunity *(foxp3*, *itgb2*, *jak3*, *lck*, *mpeg1.2*, *sike1)*, and protection against oxidative damage *(gpx3*, *gpx4b*, *gpx8*, *park7*, *prdx*, *rac2*, *selenoo1)*.

In terms of molecular processes, the GO terms activated at 24˚C were related with cardiovascular activity (*Cardiac muscle contraction*, *Hemostasis*, *Cardiac cell development*), muscle activity and development (*Myosin filament*, *Myosin complex*, *Actomyosin structure organization*), and immunity (*Macrophage activation*, *Leukocyte activation involved in inflammatory response*, *Cell killing*, *Killing of Cells of another organism*, *T-cell receptor complex*, *Lymphocyte activation*). GO categories activated at 27˚C were associated with immunity (*Leukocyte activation*, *Regulation of granulocyte differentiation*, *Macrophage activation*, *T-cell differentiation*, *Leukocyte differentiation*), apoptosis and inflammation (*Negative regulation of cell death*, *Cell death*, *Acute inflammatory response*, *Regulation of cell proliferation*), cardiovascular development (*Heart morphogenesis*, *Angiogenesis*, *Blood cell production*, *Hemopoiesis*), cellular metabolism (*Response to ketone*, *Electron transport chain*, *Cellular respiration*), muscle development (*M band*, *sarcomere*, *Troponin complex*, *Muscle cell development*, *Muscle tissue morphogenesis*), and growth (*Organ growth*, *Aging*, *Tissue development*, *Cell development*).

One of the hallmarks of thermal stress is the activation of Heat Shock Proteins (HSPs), which are highly conserved across eukaryotes and guarantee the appropriate folding of proteins in response to environmental stressors, such as high temperature. Interestingly, all comparisons between 21˚C and 27˚C for both Florida and Northern LMB showed significant activation of *hsp7c*. Further, *hspb1* was significantly activated at 27˚C at 8 and 28 DPH in the Florida LMB lineage. Two genes that were activated for both warm temperatures in Northern LMB at 28 DPH were *carhsp1* and *hspb8*. Moreover, activation of HSPs was also observed at 21˚C for Florida LMB larvae, including *hspb11* (activated at 21˚C vs. 24˚C and 27˚C at 8 DPH,

and 21°C vs. 27°C at 28 DPH), *hspa14* (21°C vs. 24°C and 27°C at 8 DPH; and 21°C vs. 27°C at 28 DPH), and *hsp90ab1* (21°C vs. 27°C at 8 and 28 DPH). Similarly, for Northern LMB larvae, *dnajc10* and *dnajc11b* showed activation at 21°C at 28 DPH.

The complete set of differentially expressed genes and enriched gene ontology categories are presented as Supplementary Data 1 and 2, and available at: https://doi.org/10.6084/m9.figshare.25620786.v1.

### 3.3. Weighted Gene Network Correlation Analyses (WGCNA)

Filtering for genes with a mean number of read counts >10 decreased the number of transcripts for the WGCNA analyses: 12,946 for Florida LMB and 17,048 for Northern LMB. For the Florida LMB, the signed clustering resulted in 10 different modules ranging from 216 genes to 2,231 (S1A Fig in S1 File). Out of these, 7 had significant associations with time to development and/or temperature treatments (Fig 5A). The Black and Blue modules had opposite trends for larvae collected at 8 and 28 DPH. The Black module was activated at 8 DPH and

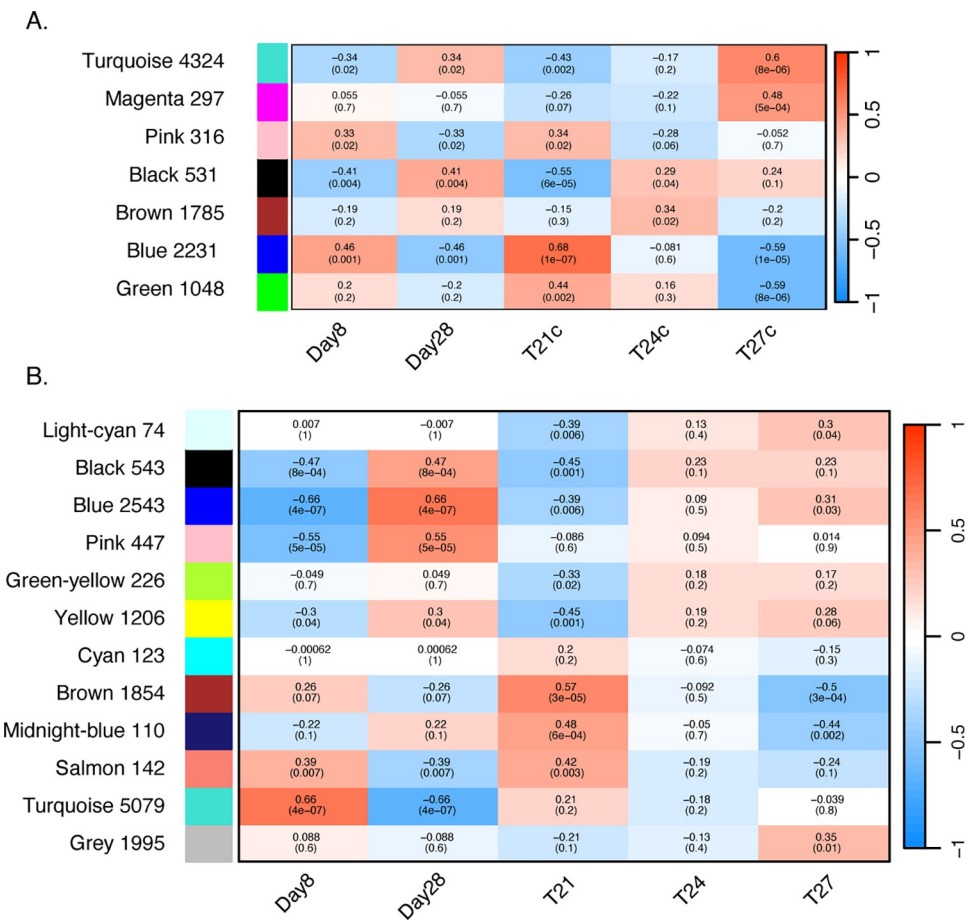

**Fig 5. Correlation between the signed modules detected with the Weighted Gene Network Correlation Analysis (WGCNA) and experimental conditions.** The heatmaps represent the two developmental stages (8- and 28-days post hatch -DPH) and three temperatures (21, 24, and 27°C) for Florida (A) and Northern (B) Largemouth Bass (LMB). The colors on the left represent the name of the correlation module, including the number of genes associated with a specific module; the color scale on the right represent the strength and direction of the association (blue negative, red positive); the numbers in the box represent the strength and direction of the association, and the numbers in parenthesis represent the adjusted p-values (significant at *p<0.01*).

deactivated at 28 DPH while showing enrichment of GO categories such as ion transport and calmodulin processing. On the other hand, the Blue module was deactivated at 8 DPH and activated at 28 DPH while showing enrichment of GO categories such as metabolism of nucleic acids, muscle development, respiratory chain complex, biosynthesis, and oxidation-reduction. There were no significant modules for individuals exposed to 24°C. Meanwhile, the modules Turquoise and Magenta showed downregulation of genes at 27°C and GO categories associated with the modules included protein synthesis, processing of nucleic acids, cell signaling, and enzyme binding. For the warmest temperature, Blue and Green modules showed a negative relationship with GO categories associated with muscle development, mitochondrial activity, electron transport chain, metabolism of amino acids, oxidation-reduction, vascular development, negative regulation of metabolism, and organization of chromatin (S2A Fig in S1 File).

For Northern LMB larvae, the WGCNA resulted in 17 different modules, which contained between 74 and 1,995 genes (S1B Fig in S1 File). Out of these, 12 had significant association with the traits of the developmental stage of the fish and temperature treatments (Fig 5B). Modules associated with stage included Black, Turquoise and Yellow, which showed opposite patterns among larvae at 8 and 28 DPH. The Black module showed enrichment for GO categories associated with muscle development, cardiovascular development, and organization of membranes, which were activated at 28 DPH. Meanwhile, modules activated at 8 DPH were Yellow (GO enrichment for electron transport, respiratory chain complex and cellular respiration) and Turquoise (enrichment for development of the nervous system). Among the modules associated with 21°C (seven in total), Light Cyan showed a negative relationship with GO categories such as cardiac and muscle development. Meanwhile, the warmest temperature (27°C) showed significant association with the modules Brown, Midnight Blue, and Grey, where the latter showed activation for RNA metabolism and processing. The strongest association was with the Brown module (70%), as shown by the correlation between module membership vs. gene significance (S2B Fig in S1 File).

## 4. Discussion

Aquaculture of freshwater species (such as LMB) is increasingly playing a key role as a reliable source of high-quality protein, especially considering the expansion of the human population in coming decades [50]. Likewise, the production of LMB for stock enhancement and recreational purposes in the U.S. is expected to expand in the near future. In this context, LMB is one of the most important freshwater species in the U.S. [18], as it is popular for food consumption and recreational angling. Several studies have shown that the physiology, morphology and behavior of fishes is controlled by both genetic and developmental conditions. Thus, it is essential to understand how different conditions during early life stages influence various genetic lineages of fishes of commercial interest. Experiments that evaluate molecular responses can promote the efficiency and sustainability of fingerling production of LMB, by providing baseline information to improve traditional rearing methods. Commercially produced LMB juveniles are usually reared in earthen ponds, which limits productivity yields due to environmental fluctuations and predation [11, 12]. Therefore, the present study utilized RAS technology, representing a viable solution for controlling rearing conditions during critical stages, and addressed questions regarding optimal temperatures for development. Moreover, temperature is one of the main factors controlling growth and development during early life stages (i.e. <30DPH), as it can produce drastic differences in aerobic metabolism, hormonal production and survival [6, 30, 51, 52]. This study investigated the potential differences in susceptibility of distinct LMB lineages to a range of developmental temperatures. In

combination with a previous study [30], these results enhance our understanding of how Florida LMB and Northern LMB respond to different thermal conditions, while showing that RAS can be used for improving survival rates during susceptible early life stages.

## 4.1. General responses to warming

In general, temperature was the dominant experimental factor for the Florida and Northern LMB lineages, as in most cases, it led to the largest differences in gene expression. These molecular observations coincide with previous results from morphological analyses, where both Florida (21°C: $\bar{x}$ = 12.04 mm, SD = 0.08; 24°C: $\bar{x}$ = 15.89 mm, SD = 0.50; 27°C $\bar{x}$ = 19.20 mm, SD = 0.63) and Northern LMB (21°C: $\bar{x}$ = 12.56, SD = 0.04; 24°C: $\bar{x}$ = 16.90, SD = 0.02; 27°C $\bar{x}$ = 19.92, SD = 0.05) showed increases in total length at warmer temperatures at 28 DPH (S1 Table in S1 File; Aguilar et al., 2023). This effect was somewhat expected, given that aquatic poikilotherms are known to develop faster in warm conditions [53, 54]. This is associated with an increase in the efficiency of chemical reactions at the cellular level, accelerating development rates and leading to observable differences in size among individuals of the same age reared at different temperatures [54]. In the present study, this was observed by the activation of genes associated with development of the nervous system and muscles for Florida LMB at 8 and 28 DPH. Meanwhile, Northern LMB at 28 DPH showed activation of genes associated with the development of muscles, cardiovascular system, bones, and nervous system, as well as cell differentiation. These patterns were confirmed with the gene module correlations of the WGCNA and the corresponding enriched GO categories.

One of the more salient genes that was activated in the comparisons of Florida LMB on 28 DPH at the warmest temperature was *poc1b*. This gene is linked to the proper formation of the cilium, which is a structure required to form cells associated with reception of external stimuli, such as light, smell, and vision [55]. In fact, inhibition of this gene in zebrafish, using morpholinos, led to body, renal, and heart defects as well as issues with retinal development [56]. In this case, it is possible that Florida LMB larvae at the warmest conditions showed higher activation of this gene due to higher growth and development rates or as activation of molecular processes associated with malformations [30].

In this regard, faster growth at elevated temperatures could be interpreted as a positive effect on survival of juvenile LMB, as it can translate to faster swimming and higher acuity for navigation, which are essential to escape predators during vulnerable life stages [57]. From a different perspective, faster growth rates at elevated temperatures can have detrimental effects on individuals and populations. At the individual level, faster development in warm conditions has been paired with shorter total lengths later in life, which could be affecting fisheries and production yields in the long term, especially as temperature increases are expected to intensify in coming decades [58, 59]. At a broader scale, faster growth can also limit the potential for dispersal in natural populations, as juvenile stages settle faster in warmer conditions [60–62]. This can reduce the genetic exchange among populations, leading to reduced genetic variability and specialization to local conditions.

## 4.2. Activation of heat-shock proteins (HSPs)

The activation of HSPs is considered one of the hallmarks of stress/repair physiology across animals, as they are stress induced chaperones associated with the appropriate folding of other proteins [63]. In the present study, despite the underlying evolutionary and ecological differences between Florida and Northern LMB, *hsp7c* was activated for all larvae at 27°C compared to those at 21°C, regardless of developmental stage. More specifically, this gene is mainly associated with the production of Heat shock cognate 71 kDa, which has been previously reported

to be activated during short term heat stress in fishes, such as salmon [64] and seabream [65]. Similarly, *hspb1* was significantly activated at 27˚C for Florida LMB larvae at both stages (8 and 28 DPH). This gene has been reported as being activated with acute warming in zebrafish embryos [66]. For Northern LMB larvae, a gene activated at both warm temperatures at 28 DPH was *hspb8*, which has been previously associated with response to acute warming in pike [67]. Both *hspb1* and *hspb8* belong to the family of small HSPs, which are associated with binding of denatured proteins and transport for cellular autophagy [68]. Given the high degree of overlap across species, these genes seem to have the potential to be effective markers of thermal stress for LMB reared in captivity and potentially also other species of commercial interest. Yet, these results remain to be confirmed via qPCR in future studies.

An interesting finding was the higher expression of some HSPs for Florida LMB larvae reared at 21˚C, including *hspb11* (activated for 21 vs. 24 and 27˚C at 8 DPH and 21 vs. 27˚C at 28 DPH), *hspa14* (21 vs. 24 and 27˚C at 8 DPH and 21 vs. 27˚C at 28 DPH), and *hsp90ab1* (21 vs. 27˚C at 8 and 28 DPH). Similarly, for Northern LMB larvae, *dnajc10* and *dnajc11b* showed higher expression at 21˚C for the 28 DPH comparisons. In this regard, we hypothesize that the higher expression of these HSPs in the colder (21˚C) "Control" conditions could be interpreted as repression of expression in larvae exposed to the warmer temperatures (24˚C and 27˚C) for extended periods. HSPs are essential during the initial stages of stress response; however, their expression is very energetically costly as they trigger several ATP-dependent processes [69, 70]. Thus, the activation of certain heat-shock proteins could be restricted to an initial (hours or days) response towards warming, after which their expression is repressed. Such patterns have also been detected in other fishes, where heatwaves of several months led to the initial activation of HSPs but were later deactivated again despite the continued presence of elevated temperatures [71]. Furthermore, in spotted seabass, the closely related *hspa14* (associated with the binding and elimination of unfolded proteins) was activated during the first three hours of warming, and expression decreased to control levels after 12 hours of exposure [72].

## 4.3. Detrimental effects of warming

At elevated temperatures, poikilotherms increase the rates of their cellular processes, demanding more energy to maintain homeostasis [3]. In our study, this was observed by the activation of GO categories associated with cellular metabolism, regulation of ATP, glucogenesis, and metabolism of lipids for Florida and Northern LMB larvae exposed to warm temperatures. Moreover, previous studies in marine and freshwater fishes have shown that increases of metabolic demands of poikilotherms exposed to warm conditions can lead to increase in oxygen consumption [4, 73]. This, in turn, has been linked to the activation of molecular pathways associated with oxygen transport and cardiac activity, with potentially hazardous side effects caused by reactive oxygen species [51, 71]. Meanwhile, Northern and Florida LMB larvae reared at 27˚C showed activation of genes relating to protection against oxidative damage, immunity, inflammation control, detoxification and apoptosis. Here, a salient example was the activation of *carhsp1* in Northern LMB larvae exposed to 24˚C and 27˚C. In mammals, this gene regulates the expression of tumor necrosis factors, which are associated with long-term inflammation and even auto-immune diseases [74, 75]. In human cells, this gene shows differential expression with temperature and is associated with the progression of viral infections [76]. Taken together, these observations indicate that both groups activated mechanisms to initiate the cellular stress/repair response due to increased metabolic demands and rates of oxygen consumption.

Even if some of the molecular pathways being activated were associated with the development of the nervous system, as seen in Florida (all developmental stages and warm

temperatures) and Northern (at 8 DPH for 24˚C) LMB, there was also a pattern for deactivation of genes corresponding to GO categories associated with neuro-transmission in Florida LMB. Similar observations have been reported in previous analyses in fishes, where brain inflammation, degeneration of neuronal mass, and oxidative damage in the brain have been described to affect the functioning of the nervous system at elevated temperatures [77]. These effects could significantly impact the capacity of fishes to learn, navigate environments, and evade predators in warm environments, ultimately affecting the survival of early life stages [78].

## 4.4. Differences between lineages

When comparing the gene expression of the larvae, it is necessary to mention the differences in diet among the Florida and Northern LMB adults, as the former were fed goldfish while the latter were fed a combination of goldfish and broodstock diet. Nutrient deficiencies and food restrictions can negatively affect fecundity, the quality of gametes, and the development of fertilized eggs [79]. However, even when dietary contrasts could have influenced the development, growth, and overall thermal response of the two lineages, the specific effects of parental diet remain unknown, and need to be evaluated in future studies. Another unexpected factor that could have influenced the gene expression results was the higher mortality reported for Florida LMB larvae reared at 27˚C (6.48% survival), compared to Northern (12.21%; 30). This would indicate that the Florida LMB larvae investigated at 28 DPH experienced a more stringent process of selection than Northern LMB, which could have a direct influence on the molecular processes.

Gene expression of Florida LMB larvae was significantly influenced by temperature (and, to a lesser degree, developmental stage), which was evidenced by the activation of genes associated with growth and development, and thermal stress for samples at 24 and 27˚C. For example, Florida LMB at 8 DPH showed several DEGs that were an order of magnitude higher than Northern individuals, suggesting a stronger influence of temperature for the former. This coincides with the finding of slightly larger total length for Florida larvae at 8 DPH for all temperatures (21˚C: $\bar{x}$ = 7.71, SD = 0.06; 24˚C: $\bar{x}$ = 8.20, SD = 0.05; 27˚C $\bar{x}$ = 8.72, SD = 0.12) when compared to Northern (21˚C: $\bar{x}$ = 7.31, SD = 0.01; 24˚C: $\bar{x}$ = 7.79, SD = 0.01; 27˚C $\bar{x}$ = 8.17, SD = 1.58; N = 100; modified from [30].

One of the key differences among the two lineages at 8 DPH was that Northern individuals showed the activation of genes related to oxygen transport, hemoglobin complex, and circulation at 24˚C and 27˚C. Since increasing temperatures lead to higher oxygen demand and higher metabolic rates in poikilotherms, activating genes associated with oxygen supply can be considered a beneficial plasticity mechanism at elevated temperatures. Moreover, Northern LMB showed the largest changes in expression at 28 DPH, showing activation of GO categories associated with growth and development of heart, muscles, and eye under warm conditions. These patterns were not as clear for Florida and suggest accelerated growth for Northern individuals at warm temperatures. These findings align with the results of the previous study by Aguilar et al. [30], which showed larger size at 28 DPH at warm temperatures for Northern (21˚C: $\bar{x}$ = 12.56, SD = 0.04; 24˚C: $\bar{x}$ = 16.90, SD = 0.02; 27˚C $\bar{x}$ = 19.92, SD = 0.05) than Florida LMB (21˚C: $\bar{x}$ = 12.04, SD = 0.08; 24˚C: $\bar{x}$ = 15.89, SD = 0.5; 27˚C $\bar{x}$ = 19.20, SD = 0.63; N = 100). Considering the combined results of the molecular and phenotypic evidence [30], Northern LMB performed better than Florida LMB in the warmer conditions. Still questions remain on how these two lineages respond to temperatures above 30˚C during early development, and how they can respond to additional stressors (e.g., acidification, high density, turbidity, etc.).

## 5. Conclusion

The present study is the first to sequence and annotate the transcriptomes of Florida and Northern LMB, constituting a valuable resource for future genetic analyses. Gene expression suggests that heat shock proteins *hspa1*, *hspa7* and *hspa8* can be informative to detect long-term heat stress in LMB. Thus, these genes could be considered as potential biomarkers for future LMB studies associated with responses to thermal fluctuations. More broadly, Florida LMB at 8 DPH exposed to warmer conditions showed an elevated number of DEGs associated with stress at the warmest temperatures, while Northern LMB at 8 DPH showed the activation of mechanisms associated with metabolic compensation. Similarly, Northern LMB individuals at 28 DPH showed large differences in expression, promoted by the activation of genes associated with growth and development. Collectively, the previous phenotypic analyses [30] and gene expression results (this study) suggest that Northern LMB is better suited for commercial production, as higher body mass and larger size were achieved, a large suite of genes associated with growth and development were activated, and higher survival than Florida LMB reared under the same conditions was demonstrated. Moreover, producers seeking to use RAS for commercial hatchery production might achieve better results than traditional earthen ponds. Still, considering some of the limitations of this study, we recommend future studies evaluate the responses of Florida and Northern LMB to broader thermal conditions across different life stages, to inform effective husbandry approaches. Techniques such as gene expression, comparative genomics and epigenetics are essential tools to understand the link between genetics and plasticity when groups of commercial interest are exposed to diverse environmental conditions during development. They are also valuable in the management of bass fisheries where knowledge of molecular responses can aide in the development of policy regarding translocation of the species outside of their native ranges. These molecular applications provide essential baseline information for groups of commercial interest, enhancing food security and sportfishing in coming decades.

## Supporting information

**S1 File.**
(DOCX)

## Acknowledgments

We would like to thank the staff of Red Hill Farms and the E.W. Shell Fisheries Center for their support with this research project. Thanks to the staff of the GSAF of UT Austin for their support with sequencing. Thanks to the support provided by the Smithsonian Tropical Research Institute and Adam Hallaj to MAB. We would like to thank the Alabama Supercomputer Authority for their assistance with bioinformatic analyses and data processing.

## Author Contributions

**Conceptualization:** Moisés A. Bernal, Anita M. Kelly, Luke A. Roy, Ian A. E. Butts.

**Data curation:** Moisés A. Bernal, Savannah L. Oglesby, Ian A. E. Butts.

**Formal analysis:** Moisés A. Bernal.

**Funding acquisition:** Moisés A. Bernal, Josh Sakmar, Allen Nicholls, Anita M. Kelly, Luke A. Roy, Ian A. E. Butts.

**Investigation:** Moisés A. Bernal, Gavin L. Aguilar, Sebastian N. Politis, Savannah L. Oglesby, Ian A. E. Butts.

**Methodology:** Gavin L. Aguilar, Josh Sakmar, Allen Nicholls.

**Project administration:** Moisés A. Bernal.

**Resources:** Josh Sakmar, Allen Nicholls.

**Supervision:** Moisés A. Bernal, Ian A. E. Butts.

**Visualization:** Moisés A. Bernal.

**Writing – original draft:** Moisés A. Bernal, Savannah L. Oglesby, Ian A. E. Butts.

**Writing – review & editing:** Moisés A. Bernal, Sebastian N. Politis, Anita M. Kelly, Luke A. Roy, Ian A. E. Butts.

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
