## [Decision Letter · Decision Letter 0]

15 Oct 2024

PONE-D-24-37286Transcriptome analyses reveal differences in the response to temperature in Florida and Northern largemouth bass (Micropterus spp.) during early life stagesPLOS ONE

Dear Dr. Bernal,

Thank you for submitting your manuscript to PLOS ONE. After careful consideration, we feel that it has merit but does not fully meet PLOS ONE’s publication criteria as it currently stands. Therefore, we invite you to submit a revised version of the manuscript that addresses the points raised during the review process.

1. This manuscript not technically sound, and the data cannot support the conclusions. PLOS ONE is designed to communicate primary scientific research, and welcome submissions in any applied discipline that will contribute to the base of scientific knowledge. But this manuscript not adhere to the criteria for scientific research article that results show not sufficient to support the conclusion.

2. This manuscript has the statistical analysis problem.

3. The revised manuscript needs to address each of the comments of the reviewers.

We look forward to receiving your revised manuscript.

Kind regards,

Tzong-Yueh Chen, Ph.D.

Academic Editor

PLOS ONE

“This project was supported by funds from Red Hill Farms. Additional support provided by Agriculture and Food Research Initiative Competitive Grant from the USDA National Institute of Food and Agriculture hatch projects 1013854 to IAEB, ALA-016-1-19075 to AMK, and ALA016-1-19053 to LAR.”

“We would like to thank the staff of Red Hill Farms and the E.W. Shell Fisheries Center for their support with this research project. Thanks to the staff of the GSAF of UT Austin for their support with sequencing. This project was supported by Agriculture and Food Research Initiative Competitive Grant from the USDA National Institute of Food and Agriculture hatch projects 1013854 to IAEB, ALA-016-1-19075 to AMK, and ALA016-1-19053 to LAR. Support was also provided by the Smithsonian Tropical Research Institute to MAB. We would like to thank the Alabama Supercomputer Authority for their assistance with bioinformatic analyses and data processing.”

“This project was supported by funds from Red Hill Farms. Additional support provided by Agriculture and Food Research Initiative Competitive Grant from the USDA National Institute of Food and Agriculture hatch projects 1013854 to IAEB, ALA-016-1-19075 to AMK, and ALA016-1-19053 to LAR.”

4. We note that Figure 1 in your submission contain copyrighted images. All PLOS content is published under the Creative Commons Attribution License (CC BY 4.0), which means that the manuscript, images, and Supporting Information files will be freely available online, and any third party is permitted to access, download, copy, distribute, and use these materials in any way, even commercially, with proper attribution. For more information, see our copyright guidelines: http://journals.plos.org/plosone/s/licenses-and-copyright.

Reviewers' comments:

Reviewer's Responses to Questions

**Comments to the Author**

1. Is the manuscript technically sound, and do the data support the conclusions?

Reviewer #1: Partly

Reviewer #2: Yes

2. Has the statistical analysis been performed appropriately and rigorously? 

Reviewer #1: Yes

Reviewer #2: I Don't Know

3. Have the authors made all data underlying the findings in their manuscript fully available?

Reviewer #1: Yes

Reviewer #2: Yes

4. Is the manuscript presented in an intelligible fashion and written in standard English?

Reviewer #1: Yes

Reviewer #2: Yes

5. Review Comments to the Author

Reviewer #1: In general, this is an interesting and well-organized manuscript. The authors sequenced and annotated the transcriptome from two different species of largemouth bass larvae in responding to divergent temperatures. The resulting differential expression profiles reveal some genetic aspects that reflect changes of temperature at early development stages of LMBs. Here, I listed some questions relevant to experimental design and data analysis. If the authors can adequately address these questions in the revised manuscript, I will suggest to accepting the revision for publication.

1. According to the preliminary studies (line 81 - 82), 27 oC can be seeing as the optimal growing temperature for LMBs and their range of tolerating is quite large. So, my question is what’s the reason that three experimental temperatures (21, 24 and 27 oC) were selected? Why is that only colder (21 and 24 oC) temperatures were examined as control (21 oC) and intermediate (24 oC) in the current study? Do you have plan to evaluate warmer living conditions (>28 oC) for LMBs?

2. The authors defined critical early life stages as <30-days post hatch (line 98), and therefore, larvae at 8-DPH and 28-DPH were selected to be examined in the present study. Do you have any literature evidence to support this notion that <30-DPH is significant to larval development and/or critical to aquacultural husbandry, which considering as justifying the experimental design?

3. Is there any specific reason for the authors to use Trinity to assembling transcriptomic sequences (line 200)? Since Trinity is known to have problems handling lowly expressed transcripts and high coverage regions, and its normalization process is also questionable (Honaas et al., 2016) and (https://github.com/trinityrnaseq/trinityrnaseq/wiki/Trinity-FAQ).

4. Although the authors summarized the experimental conditions with fish treatments in Table 1, there is no detail information stating how many samples were sequenced and how the technical and biological repeats were generated. Please provide this information for further evaluation of the reliability of experiment. In addition, do you pool tissue? If not, how do you generalize the bias between individuals?

5. In regarding to determining the differential expression genes (DEGs), since there are two variables included in the experimental design (temperatures and age: DPH), I will need the authors to clarify how to normalize the sequencing data into relative abundances of each mRNA in representing relative expression levels of each gene?

6. The qualities of picture and diagram in Figure 2, 3, and 5 are poor. Please revise these figures with better quality.

7. For Figure 5, What does the color bar indicate to? Does the red color indicating the degree of positively correlated and blue color indicating the degree of negatively correlated to certain GO terms? Clarification and a more meaningful figure legend are required.

Reviewer #2: The manuscript (PONE-D-24-37286) presents a technically sound study using transcriptomic data to explore how two lineages of largemouth bass respond to temperature variations during early life stages. The data is well-supported by the methodology, and the conclusions regarding differential gene expression under various temperature conditions are valid based on the presented results. The inclusion of gene ontology (GO) and weighted gene co-expression network analysis (WGCNA) enriches the interpretation of the functional significance of differentially expressed genes (DEGs). However, to ensure the completeness of the content, there are some areas that could be improved:

The introduction provides a solid overview of the importance of temperature regulation in early fish development. However, the link between the specific hypotheses tested in this study and previous research on largemouth bass transcriptomics could be made clearer. I recommend adding more detail on how this study builds upon existing work in this field, specifically emphasizing the novelty of comparing the two largemouth bass lineages (Florida and Northern). This addition would provide a stronger foundation for the reader to understand how the study advances the current knowledge on fish transcriptomics and thermal adaptation.

The description of the experimental design could benefit from further elaboration on the triplicate data used for each treatment group. The manuscript currently lacks sufficient detail on the replication strategy for the two largemouth bass lineages (Florida and Northern) across the three temperature conditions. Please clarify how the triplicates were incorporated into the experimental setup and any statistical methods used to account for this in the analysis. This is essential for reproducibility and understanding the robustness of the data.

Page 15: In Table 2, which presents the contig statistics and BUSCO results for the Florida and Northern largemouth bass transcriptomes, there is a notable difference in the "Fragmented BUSCOS" category. Specifically, the Florida largemouth bass transcriptome has 113 fragmented BUSCOs (3.1%) compared to 83 fragmented BUSCOs (2.3%) in the Northern transcriptome. This discrepancy warrants further discussion. Could this difference be due to variations in sequencing quality or depth between the two lineages, or might it reflect underlying biological factors such as genetic divergence between the populations? I recommend addressing this point in the manuscript, perhaps by providing additional details on the assembly process or discussing potential biological explanations. Clarifying this difference would enhance the readers' understanding of the assemblies' quality and their implications for downstream analyses.

Page 17: Figure 4. There appears to be an inconsistency between Figure 4 and its caption. The caption refers to comparisons across temperature treatments at different developmental stages for both Florida (A) and Northern (B) largemouth bass, but it also implies the existence of panels (C) and (D), which are either missing or mislabelled. To ensure clarity and accurate representation of the data, I suggest revising the figure to include the missing panels or adjusting the caption to match the figure as currently presented. Consistent labeling of figures and captions is essential for clear communication of the results.

The presentation style of Table 3 resembles more of a diagram than a traditional table, which may lead to confusion. For better clarity and consistency with the rest of the manuscript, I recommend reformatting Table 3 into a standard table layout with clear rows and columns. This will make the data more accessible and easier to interpret for the reader, thereby improving the overall presentation of the results.

Figure 5 is well-structured, but there are areas where the caption and its reference in the manuscript can be improved. Specifically:

1) Please use consistent terminology for "Weighted Gene Network Correlation Analysis" (WGCNA). The abbreviation WGCA is also mentioned elsewhere—ensure uniformity throughout the manuscript.

2) Clarify how correlation strength and statistical significance are represented in the figure. The description of the correlation coefficients and p-values could be more transparent for readers unfamiliar with the methodology.

3) Ensure that the manuscript provides a thorough biological interpretation of the correlations shown in Figure 5, particularly highlighting key findings and their implications for understanding gene expression in response to temperature across developmental stages.

The discussion section interprets the findings effectively within the broader context of thermal tolerance in fish. However, it would benefit from a deeper exploration of how these transcriptomic changes might affect long-term survival, growth, and reproduction in the wild. Additionally, there are opportunities to discuss the potential applications of these findings in management, conservation, and aquaculture. For instance, how might selective breeding programs leverage these results to improve thermal tolerance in largemouth bass populations? Expanding on these points could provide greater relevance to the study's broader ecological and practical implications.

6. PLOS authors have the option to publish the peer review history of their article (what does this mean?). If published, this will include your full peer review and any attached files.

Reviewer #1: No

Reviewer #2: No

---

## [Author Response · Author response to Decision Letter 0]

13 Dec 2024

Responses to reviewer’s comments: 

Editorial comments: 

Figure 1 has been modified to avoid any copyright issues. The photographs were taken by two co-authors of the study. Further, all the additional information for bioinformatic pipelines and data availability is now freely accessible to the public. 

We would like to modify the statement of financial disclosure, based on the comments of the editor: 

“This project was supported by funds from Red Hill Farms. Additional support provided by Agriculture and Food Research Initiative Competitive Grant from the USDA National Institute of Food and Agriculture hatch projects 1013854 to IAEB, ALA-016-1-19075 to AMK, and ALA016-1-19053 to LAR. The funders had no role in study design, decision to publish, or preparation of the manuscript."”

Reviewer #1: 

In general, this is an interesting and well-organized manuscript. The authors sequenced and annotated the transcriptome from two different species of largemouth bass larvae in responding to divergent temperatures. The resulting differential expression profiles reveal some genetic aspects that reflect changes of temperature at early development stages of LMBs. Here, I listed some questions relevant to experimental design and data analysis. If the authors can adequately address these questions in the revised manuscript, I will suggest to accepting the revision for publication.

1. According to the preliminary studies (line 81 - 82), 27 oC can be seeing as the optimal growing temperature for LMBs and their range of tolerating is quite large. So, my question is what’s the reason that three experimental temperatures (21, 24 and 27 oC) were selected? Why is that only colder (21 and 24 oC) temperatures were examined as control (21 oC) and intermediate (24 oC) in the current study? Do you have plan to evaluate warmer living conditions (>28 oC) for LMBs?

Comments: Thanks for the careful observation. The challenge with the study of Large-mouth bass is that this is a species complex, and older studies do not make a distinction of the different groups being analyzed. Further, as reported in previous studies in ectotherms, developing at warmer temperatures can affect the thermal preferences and tolerance. For example, several studies have been conducted in Mexico, where the thermal preference and tolerance appears to be higher than in other latitudes. The LMB we are analyzing in this study corresponds to Northern and Florida that traditionally breed at temperatures between 20C and 22C, which means temperatures of 24C and 27C are expected to produce some challenges. This is highlighted in lines 79-83 of the introduction, and 202-207 of the methods. 

2. The authors defined critical early life stages as <30-days post hatch (line 98), and therefore, larvae at 8-DPH and 28-DPH were selected to be examined in the present study. Do you have any literature evidence to support this notion that <30-DPH is significant to larval development and/or critical to aquacultural husbandry, which considering as 

justifying the experimental design?

Response: Thanks for the observation. In general, for aquatic organisms this is the time when they are most sensitive to environmental fluctuations and experience the highest levels of mortality. Specifically for fish husbandry, this is the time where the largest changes in morphology occur, and they undergo transitions in food supplies. All this combined makes them some of the most challenging periods in the lifetime of a fish. 

We added lines 105-108 to highlight this: “Despite these advances, questions remain on how temperature influences underlying molecular processes in Florida and Northern LMB during the “critical” early life stages (i.e. <30DPH) when fishes undergo drastic changes in food sources, morphology, and elevated mortality (33).”

3. Is there any specific reason for the authors to use Trinity to assembling transcriptomic sequences (line 200)? Since Trinity is known to have problems handling lowly expressed transcripts and high coverage regions, and its normalization process is also questionable (Honaas et al., 2016) and (https://github.com/trinityrnaseq/trinityrnaseq/wiki/Trinity-FAQ).

Response: Thanks for the observation. The program Trinity has been used extensively for assembling de novo transcriptomes of non-model species. The original manuscript has been cited >8,000 times (Hass et al. 2013, Nature Protocols) in a broad range of taxa. Despite some of its limitations, it is still considered reliable software and widely used for this type of analyses. For example, the manuscript “Hölzer, M., & Marz, M. (2019). De novo transcriptome assembly: A comprehensive cross-species comparison of short-read RNA-Seq assemblers. Gigascience, 8(5)” compares most available de novo transcriptome methods and they highlight that Trinity is one of the most useful and reliable methods available. Finally, a direct comparison of transcriptome or genome assembly methods is beyond the scope of the presented study. 

4. Although the authors summarized the experimental conditions with fish treatments in Table 1, there is no detail information stating how many samples were sequenced and how the technical and biological repeats were generated. Please provide this information for further evaluation of the reliability of experiment. In addition, do you pool tissue? If not, how do you generalize the bias between individuals?

Response: As described in the methods, RNA extractions were conducted for whole individual larvae, and none of the samples were pooled. Moreover, the larvae included in the analyses were selected from different tanks, as shown in Table 1. The Likelihood-Ratio Test with DESEq2 suggests that tank had a very minor effect on the number of differentially expressed genes. This explanation has been expanded in lines 281-305: 

“Individual larvae were extracted using the Quick-RNA tissue kit (Zymo, Irvine, CA, USA), following the manufacturer’s protocols. The only modifications of the protocol were the time to incubation of samples with Proteinase K (overnight at 32°C) and the adjustment of volume for the final elution to 60 µL. Individual larvae extractions that had a concentration higher than 100 ng/µL of RNA, and an integrity (RIN) of 6.5 or above were used for library preparation. The library preparation and sequencing were completed for individual larvae, to avoid any biases from pooling RNA from different samples. TagSeq library preparation and sequencing were conducted at the GSAFU of the University of Texas in Austin (USA), following the protocol delineated by Meyer et al. (43,44). Previous studies indicate that Tag-Seq is a very reliable method for assessing differential expression, showing strong correlation with control mRNA genes (43,45). Sequencing was performed with an Illumina Hiseq4000, and fragments were 50 bp single end. Raw reads for the gene expression reads are available through NCBI BioProject PRJNA1100459 (SUB14448778).”

5. In regarding to determining the differential expression genes (DEGs), since there are two variables included in the experimental design (temperatures and age: DPH), I will need the authors to clarify how to normalize the sequencing data into relative abundances of each mRNA in representing relative expression levels of each gene?

Response: thanks for the observation, lines 351-354 of the methods now state: “The RNAseq reads were normalized following the standard DESeq2 method of median-of-ratios, where the main factors are the sequencing depth for specific genes compared across the different individual conditions (thermal treatments and DPH).”

6. The qualities of picture and diagram in Figure 2, 3, and 5 are poor. Please revise these figures with better quality.

Response: Thanks for the observation, the resolution of the images has been improved. 

7. For Figure 5, What does the color bar indicate to? Does the red color indicating the degree of positively correlated and blue color indicating the degree of negatively correlated to certain GO terms? Clarification and a more meaningful figure legend are required.

Response: This has been modified for clarity and it now reads: “Fig 5. Correlation between the signed modules detected with the Weighted Gene Network Correlation Analysis (WGCNA) and experimental conditions. The heatmaps represent the two developmental stages (8- and 28-days post hatch -DPH) and three temperatures (21, 24, and 27°C) for Florida (A) and Northern (B) Largemouth Bass (LMB). The colors on the left represent the name of the correlation module, including the number of genes associated with a specific module; the color scale on the right represent the strength and direction of the association (blue negative, red positive); the numbers in the box represent the strength and direction of the association, and the numbers in parenthesis represent the adjusted p-values (significant at p<0.01). ”

Reviewer #2: 

The manuscript (PONE-D-24-37286) presents a technically sound study using transcriptomic data to explore how two lineages of largemouth bass respond to temperature variations during early life stages. The data is well-supported by the methodology, and the conclusions regarding differential gene expression under various temperature conditions are valid based on the presented results. The inclusion of gene ontology (GO) and weighted gene co-expression network analysis (WGCNA) enriches the interpretation of the functional significance of differentially expressed genes (DEGs). However, to ensure the completeness of the content, there are some areas that could be improved:

1. The introduction provides a solid overview of the importance of temperature regulation in early fish development. However, the link between the specific hypotheses tested in this study and previous research on largemouth bass transcriptomics could be made clearer. I recommend adding more detail on how this study builds upon existing work in this field, specifically emphasizing the novelty of comparing the two largemouth bass lineages (Florida and Northern). This addition would provide a stronger foundation for the reader to understand how the study advances the current knowledge on fish transcriptomics and thermal adaptation.

Response: Thanks for the observation. Based on this comment and suggestions from Reviewer 1 we have revised this section extensively. Lines 78-83 now read: “Fewer studies have focused on the effects of temperature on early development. This requires careful analyses given that the variable developmental conditions experienced early in life can lead to long-term effects of thermal tolerance and temperature preference (26). Further, members of the LMB complex have been introduced to different locations globally, and the diverse genetic lineages of LMB available for commercial husbandry have not been specified in previous studies.”

Lines 114-116 now read: “Previous studies have reported significant genetic differences between these two groups (34,35), but in many cases, reports do not specify the genetic lineage that is being evaluated.”

Lines 140-145 now read: “Considering that fish at higher latitudes tend to develop faster as they have shorter growing seasons (i.e., counter gradient growth hypothesis; 37), we hypothesize that Northern LMB will have faster growth than Florida LMB, and that this will be reflected in the activation of molecular pathways associated with development. By expanding the genomic resources for two lineages of Micropterus spp., this study represents an essential contribution to improving LMB aquaculture worldwide.”

2. The description of the experimental design could benefit from further elaboration on the triplicate data used for each treatment group. The manuscript currently lacks sufficient detail on the replication strategy for the two largemouth bass lineages (Florida and Northern) across the three temperature conditions. Please clarify how the triplicates were incorporated into the experimental setup and any statistical methods used to account for this in the analysis. This is essential for reproducibility and understanding the robustness of the data.

Response: Thanks for the observation, this has been edited extensively for clarity. 

Lines 172-174 read: “Florida broodstock comprised 34 males and 40 females (length: 321 to 584 mm; weight: 0.48 to 3.97 kg), while Northern broodstock comprised 74 males and 56 females (length: 330 to 481 mm; weight: 0.56 to 1.99 kg).”

Lines 202-223 read: “Three temperature treatments (Control = 21°C, Intermediate = 24°C, or Warm = 27°C) were chosen based on the temperatures experienced by the spawning adults (19°C-23°C) and the temperatures experienced during the egg incubation (21°C). Each one of the thermal treatments were setup in one RAS (i.e., three separate RAS total), and the larvae used for the gene expression analyses come from four to six different tanks within each recirculating system (Table 1). The RAS were equipped with a UV sterilizer, bead and bag filters, heaters, and chillers for temperature control (30). Temperature and DO were checked twice daily with a YSI multiparameter meter (YSI, Yellow Springs, OH, USA), and the temperature fluctuation was minimal (± 0.2°C).”

Further Table 1 has been edited for clarity. 

3. Page 15: In Table 2, which presents the contig statistics and BUSCO results for the Florida and Northern largemouth bass transcriptomes, there is a notable difference in the "Fragmented BUSCOS" category. Specifically, the Florida largemouth bass transcriptome has 113 fragmented BUSCOs (3.1%) compared to 83 fragmented BUSCOs (2.3%) in the Northern transcriptome. This discrepancy warrants further discussion. Could this difference be due to variations in sequencing quality or depth between the two lineages, or might it reflect underlying biological factors such as genetic divergence between the populations? I recommend addressing this point in the manuscript, perhaps by providing additional details on the assembly process or discussing potential biological explanations. Clarifying this difference would enhance the readers' understanding of the assemblies' quality and their implications for downstream analyses.

Response: Thanks for the observation. Usually these differences in completeness, average contig length, and numbers of fragmented BUSCOs are based on unpredictable factors during library preparation and sequencing. As described by Kim et al. 2022 (Scientific Reports) these two species are phylogenetically closely related, thus, we don’t expect these differences to be produced by chromosomal rearrangements or differences in genomic structure. 

This is now presented in lines 403-407: “Differences among the two denovo transcriptomes based on the number of contigs, or the number of complete or fragmented BUSCOs are most likely associated with stochastic factors such as the effectiveness of sequencing and assembly. The two species are closely related and still capable of hybridizing (35), thus we do not expect that these differences are associated with differences in chromosomal rearrangements or genome duplications.” 

4. Page 17: Figure 4. There appears to be an inconsistency between Figure 4 and its caption. The caption refers to comparisons across temperature treatments at different developmental stages for both Florida (A) and Northern (B) largemouth bass, but it also implies the existence of panels (C) and (D), which are either missing or mislabelled. To ensure clarity and accurate representation of the data, I suggest revising the figure to include the missing panels or adjusting the caption to match the figure as currently presented. Consistent labeling of figures and captions is essential for clear communication of the results.

Response: Thanks for the careful observation. The legend has been edited and it now reads: 

“Fig 4. Venn diagrams showing the number of Differentially Expressed Genes. The comparisons were among different temperature treatments (21°C, 24°C, and 27°C) at different developmental stages (8- and 28-days post hatch- DPH) for Florida (A- 8DPH; B- 28DPH) and Northern (C- 8DPH; D 28DPH) Largemouth Bass (LMB) larvae.”

5. The presentation style of Table 3 resembles more of a diagram than a traditional table, whi

---

## [Decision Letter · Decision Letter 1]

2 Jan 2025

Transcriptome analyses reveal differences in the response to temperature in Florida and Northern Largemouth Bass (Micropterus spp.) during early life stages

PONE-D-24-37286R1

Dear Dr. Bernal,

We’re pleased to inform you that your manuscript has been judged scientifically suitable for publication and will be formally accepted for publication once it meets all outstanding technical requirements.

Kind regards,

Tzong-Yueh Chen, Ph.D.

Academic Editor

PLOS ONE

Additional Editor Comments (optional):

Reviewers' comments:

Reviewer's Responses to Questions

**Comments to the Author**

1. If the authors have adequately addressed your comments raised in a previous round of review and you feel that this manuscript is now acceptable for publication, you may indicate that here to bypass the “Comments to the Author” section, enter your conflict of interest statement in the “Confidential to Editor” section, and submit your "Accept" recommendation.

Reviewer #1: All comments have been addressed

Reviewer #2: All comments have been addressed

2. Is the manuscript technically sound, and do the data support the conclusions?

Reviewer #1: Yes

Reviewer #2: Yes

3. Has the statistical analysis been performed appropriately and rigorously? 

Reviewer #1: Yes

Reviewer #2: Yes

4. Have the authors made all data underlying the findings in their manuscript fully available?

Reviewer #1: Yes

Reviewer #2: Yes

5. Is the manuscript presented in an intelligible fashion and written in standard English?

Reviewer #1: Yes

Reviewer #2: Yes

6. Review Comments to the Author

Reviewer #1: Since the authors adequately addressed my questions and revised the text and figures accordingly, I will recommend to accept the manuscript for publication.

Reviewer #2: The authors have addressed the reviews individually and provided valuable insights into the field. The manuscript is now suitable for publication in its current form.

7. PLOS authors have the option to publish the peer review history of their article (what does this mean?). If published, this will include your full peer review and any attached files.

Reviewer #1: No

Reviewer #2: No

---

## [Editor Report · Acceptance letter]

6 Jan 2025

PONE-D-24-37286R1 

PLOS ONE

Dear Dr. Bernal, 

I'm pleased to inform you that your manuscript has been deemed suitable for publication in PLOS ONE. Congratulations! Your manuscript is now being handed over to our production team.

Kind regards, 

on behalf of

Prof. Tzong-Yueh Chen 

Academic Editor

PLOS ONE
